# Quantification of impact of COVID-19 pandemic on cancer screening programmes – a case study from Argentina, Bangladesh, Colombia, Morocco, Sri Lanka, and Thailand

Eric Lucas[1], Raul Murillo[2], Silvina Arrossi[3], Martin Bárcena[4], Youssef Chami[5], Ashrafun Nessa[6], Suraj Perera[7], Padmaka Silva[7], Suleeporn Sangrajrang[8], Richard Muwonge[1], Partha Basu[1]*

[1]International Agency for Research on Cancer, Lyon, France; [2]Hospital Universitario San Ignacio, Bogota, Colombia; [3]Investigadora CEDES/CONICET, Buenos Aires, Argentina; [4]Instituto Provincial del Cáncer, Jujuy, Argentina; [5]Foundation Lalla Salma Cancer prevention and treatment, Rabat, Morocco; [6]Bangabandhu Sheikh Mujib Medical University, Dhaka, Bangladesh; [7]Ministry of Health, Colombo, Sri Lanka; [8]National Cancer Institute of Thailand, Bangkok, Thailand

*For correspondence:
basup@iarc.who.int

Competing interest: The authors declare that no competing interests exist.

**Abstract** It is quite well documented that the COVID-19 pandemic disrupted cancer screening services in all countries, irrespective of their resources and healthcare settings. While quantitative estimates on reduction in volume of screening tests or diagnostic evaluation are readily available from the high-income countries, very little data are available from the low- and middle-income countries (LMICs). From the CanScreen5 global cancer screening data repository we identified six LMICs through purposive sampling based on the availability of cancer screening data at least for the years 2019 and 2020. These countries represented those in high human development index (HDI) categories (Argentina, Colombia, Sri Lanka, and Thailand) and medium HDI categories (Bangladesh and Morocco). No data were available from low HDI countries to perform similar analysis. The reduction in the volume of tests in 2020 compared to the previous year ranged from 14.1% in Bangladesh to 72.9% in Argentina (regional programme) for cervical screening, from 14.2% in Bangladesh to 49.4% in Morocco for breast cancer screening and 30.7% in Thailand for colorectal cancer screening. Number of colposcopies was reduced in 2020 compared to previous year by 88.9% in Argentina, 38.2% in Colombia, 27.4% in Bangladesh, and 52.2% in Morocco. The reduction in detection rates of CIN 2 or worse lesions ranged from 20.7% in Morocco to 45.4% in Argentina. Reduction of breast cancer detection by 19.1% was reported from Morocco. No association of the impact of pandemic could be seen with HDI categories. Quantifying the impact of service disruptions in screening and diagnostic tests will allow the programmes to strategize how to ramp up services to clear the backlogs in screening and more crucially in further evaluation of screen positives. The data can be used to estimate the impact on stage distribution and avoidable mortality from these common cancers.

## Editor's evaluation

This study provides important estimates from an international cancer screening data repository about the impact of the COVID-pandemic related disruptions on cancer screening programs in selected low- and middle-income countries. The evidence supporting the study is solid and relies

on national-level screening program attendee volumes and assessments of screen positives during 2019 (pre-pandemic) and 2020 (during the pandemic). The study provides real-world data estimates of proportions/volumes of missed screenings due to pandemic control measures (lockdowns and closures) and may contribute to future modelling efforts for measuring the impact on late/advanced stage detection and excess case burden and mortality.

## Introduction

Breakout of the coronavirus pandemic in 2019 with the emergence of the SARS-Cov-2 virus severely impacted the ability of countries to address and respond to the growing burden of cancers. Mitigation measures such as lockdowns, social distancing norms, and travel restrictions along with closure of non-emergency health services disrupted cancer care and imposed access delays (*WHO, 2020a*). The impact of the COVID-19 pandemic on cancer screening, symptomatic diagnosis of cancer, referral pathways, and cancer management has been well documented in high-resourced countries (*Jones et al., 2020*). In the United States of America, a significant drop in the volume of screening for breast cancer (94%), colon cancer (86%), and cervical cancer (94%) was reported between January 2020 and April 2020 (*Epic Health Research Network, 2020*). However, such quantitative estimates are rarely available from outside of North America, Australia, or Europe.

A cross-sectional survey was conducted in 2020 by the International Agency for Research on Cancer (IARC), France to assess the impact of COVID-19 pandemic on cancer screening programmes in 17 countries (*Villain et al., 2021*). Though the survey documented country experiences with programme suspensions, changes in health priorities, and curtailment of follow-up visits, it was too early to generate any quantitative estimate on their impact on screening and downstream processes. In the present article, we have reported impact of the COVID-19 pandemic on existing cancer screening programmes in selected low- and middle-income countries (LMICs) based on comparison of volume of screening and further assessment of screen positives, before and during the pandemic. Sampling of the countries/programmes was purposive, based on their ability to share cancer screening performance data covering the years before and during the pandemic. We included countries belonging to different categories of human development index (HDI) in our purposive sampling of LMICs to document if impact of the pandemic was different among them.

## Methods

The IARC-led project 'Cancer Screening in Five Continents (CanScree5)' is a global repository of information on cancer screening programmes, which has collected cancer screening information and data from 84 countries (including LMICs) till date (*Basu et al., 2019*). LMICs reported to be collecting screening performance data on yearly basis were selected out of the participating countries. Among these, we invited focal points of cancer screening programmes from two very high HDI category LMICs (Argentina and Thailand; HDI between 0.8 and 1.0 in 2020), two high HDI category LMICs (Colombia and Sri Lanka; HDI between 0.7 and 0.79 in 2020), and two medium HDI category LMICs (Bangladesh and Morocco; HDI between 0.55 and 0.69). The cancer screening programme focal points from these countries also informed us that they had data available at least for the years 2019 (pre-pandemic) and 2020 (during the pandemic). None of the programmes from low HDI countries participating in CanScreen5 project had quantitative data that could be shared with us.

The focal point(s) from each country shared available data on number screened, and number further assessed disaggregated by months or quarters for the years 2019 and 2020 to allow direct comparison. All the countries except Thailand reported for cervical screening programmes. Data were obtained from breast cancer screening programmes in Bangladesh, Colombia, Morocco, and Sri Lanka. Colorectal cancer screening programme data were reported from Thailand only. All the countries except Argentina submitted national data. Colombia submitted national data obtained only from two insurance companies representing 23.8% of the total population. Cancer screening data from Morocco were disaggregated by regions and not by months. The qualitative information on the screening protocol and programme organization was collected from the CanScreeen5 website (*Basu et al., 2019*).

For each country, we have summarized the protocol and organization of the cancer screening programme, screening performance for the years 2019 and 2020, and evolution of COVID-19 burden based on data available at the WHO COVID-19 dashboard (*WHO, 2020b*). Due to the diversity of protocol, screening tests, referral pathway adopted by the countries and the different information systems used, the data across the countries are not directly comparable. Hence, we have presented the outcomes as case studies from the six target countries.

The CanScreen5 project was reviewed by IARC Ethic Committee and was approved for collection of programme information and aggregate data on an ongoing basis.

## Results
### Case study 1: Argentina (very high HDI category) (Jujuy province)

Argentina is a federal country divided into 24 provinces. Though the National Ministry of Health provides broad regulatory framework and guidelines, the health programmes are administered at provincial level (*Arrossi et al., 2021*). The National Program on Cervical Cancer Prevention (NPCCP) was relaunched in 2008 targeting women between 25 and 65 years of age (*Arrossi et al., 2015*). HPV testing was introduced in Jujuy through the Jujuy Demonstration project, led by the National Program on Cervical Cancer Prevention. HPV testing is offered to women aged 30 years and older who attend the public health system. Cytology is used as the triage test (*Arrossi et al., 2019*). There is no invitation system for the entire target population. Community health workers from Jujuy have a nominalized list of women with public health insurance eligible to be screened. This list is used to offer HPV-self collection during their routine visits of households or invite women to have HPV testing at health centres. This system is limited to socio-economically disadvantaged women only. The cervical screening coverage in Argentina was reported to exceed 70% of the eligible *population* (*ICO/IARC, 2021a*; *WHO, 2021a*).

**Table 1.** Monthly change in number of women undergoing screening, colposcopy, and/or treatment, and high-grade CIN detected in the cervical cancer screening programme in Jujuy province, Argentina during 2019 and 2020.

| Year of screening | Month of screening | | | | | | | | | | | | Total |
|---|---|---|---|---|---|---|---|---|---|---|---|---|---|
| | Jan | Feb | Mar | Apr | May | Jun | Jul | Aug | Sep | Oct | Nov | Dec | |
| *Number of women screened (with cytology or HPV test)* | | | | | | | | | | | | | |
| 2019 – n | 4128 | 4140 | 4457 | 5726 | 5969 | 4036 | 4579 | 5200 | 6205 | 5973 | 5319 | 2611 | 58,343 |
| 2020 – n | 3102 | 2577 | 1312 | 291 | 840 | 875 | 214 | 189 | 295 | 1351 | 2608 | 2169 | 15,823 |
| Change – %* | −24.9 | −37.8 | −70.6 | −94.9 | −85.9 | −78.3 | −95.3 | −96.4 | −95.2 | −77.4 | −51.0 | −16.9 | −72.9 |
| *Number of women undergoing colposcopy* | | | | | | | | | | | | | |
| 2019 – n | 234 | 272 | 322 | 374 | 314 | 240 | 208 | 378 | 433 | 356 | 318 | 305 | 3,754 |
| 2020 – n | 62 | 49 | 37 | 37 | 60 | 43 | 13 | 11 | 18 | 20 | 20 | 45 | 415 |
| Change – %* | −73.5 | −82.0 | −88.5 | −90.1 | −80.9 | −82.1 | −93.8 | −97.1 | −95.8 | −94.4 | −93.7 | −85.2 | −88.9 |
| *Number of women detected with CIN 2 or worse lesions* | | | | | | | | | | | | | |
| 2019 – n | 23 | 34 | 28 | 41 | 27 | 29 | 18 | 42 | 45 | 30 | 51 | 31 | 399 |
| 2020 – n | 35 | 26 | 15 | 19 | 24 | 21 | 8 | 8 | 10 | 15 | 13 | 24 | 218 |
| Change – %* | +52.2 | −23.5 | −46.4 | −53.7 | −11.1 | −27.6 | −55.6 | −81.0 | −77.8 | −50.0 | −74.5 | −22.6 | −45.4 |
| *Number of LLETZ or conization performed* | | | | | | | | | | | | | |
| 2019 – n | 9 | 12 | 13 | 30 | 25 | 16 | 16 | 18 | 18 | 18 | 24 | 20 | 219 |
| 2020 – n | 18 | 10 | 8 | 4 | 19 | 14 | 5 | 4 | 8 | 9 | 18 | 9 | 126 |
| Change – %* | +100.0 | −16.7 | −38.5 | −86.7 | −24.0 | −12.5 | −68.8 | −77.8 | −55.6 | −50.0 | −25.0 | −55.0 | −42.5 |

*Change is calculated as (numbers in 2020 − numbers in 2019)/(numbers in 2019) × 100; HPV: human papilloma virus; CIN: cervical intraepithelial neoplasia; LLETZ: large loop excision of the transformation zone.

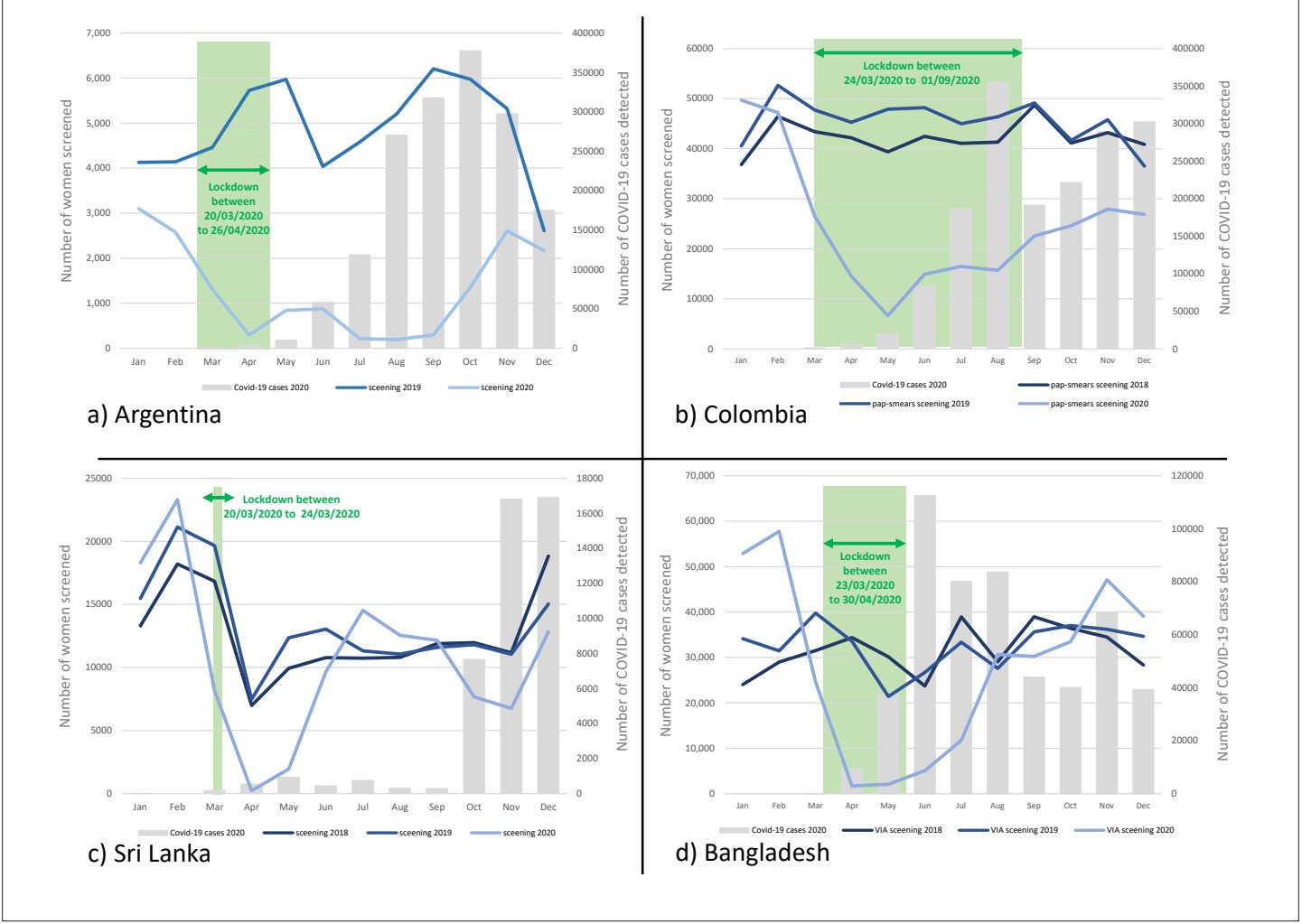

**Figure 1.** Number of women screened for cervical cancer in Argentina (Jujuy province only) (**a**), Colombia (**b**), Sri Lanka (**c**), and Bangladesh (**d**) by months in selected years; the bars show the number of COVID-19 cases detected by months in 2020; the green bar shows the lockdown period.

In Argentina, an online information system (SITAM, for its initial in Spanish) is used to record data for cervical cancer screening and downstream management. The system and its technical support and maintenance are funded by the national Ministry of Health (***Drummond et al., 2017***).

Community transmission of SARS-Cov-2 in Argentina was officially declared on 3 March 2020. As of 25 January 2023, total 10.0 million confirmed cases of COVID-19 and 130,394 deaths (0.35% of population) were reported (***WHO, 2020c***). Health authorities enforced first national lockdown between 20 March and 26 April 2020. However, partial lockdowns for local containment of disease were implemented until 30 August 2020. However, by September–October 2020 Jujuy became one of the Argentinean provinces with highest cumulative infection and mortality rates. During this acute phase of disease spread governmental measures focused mainly on preventing the collapse of the health system. By August–September, 93% of Jujuy intensive care beds were occupied with COVID-19 cases. Interruptions in provision of primary healthcare and problems to access drugs were reported (***Bernasconi et al., 2021***).

In the JuJuy province, total number of women screened with either cytology or HPV test in a year dropped by 72.9% in 2020 from a total of 58,343 women screened in the year 2019 (***Table 1***). The impact was most severe during the lockdown period and its immediate aftermath with highest reduction of 96.4% being reported in the month of August 2020. With lifting of sanitary restrictions, a gradual improvement of the number of women screened per month was observed only during the year end (***Figure 1a***). The recovery was slow due to the persistent high burden of COVID cases.

Number of colposcopies performed on the screen-positive women in 2020 was also drastically reduced by 88.9% (*Table 1*). The overall proportion of screened women undergoing colposcopy in 2019 was 6.4% with the proportion remaining similar across the months. The proportion was reduced to only 2.6% in 2020, highlighting the large number of screen-positive women missing their further assessment and possible treatment. Due to smaller number of women being screened and smaller proportion of screen-positive women being assessed with colposcopy in 2020, there was 45.4% reduction in the number of cervical intraepithelial neoplasia 2 or worse disease (CIN 2+) being detected, and 42.5% excisional treatment being done in the same year compared to 2019.

## Case study 2: Colombia (very high HDI category)

Colombia introduced cervical and breast cancer screening programmes in the year 2000. The screening protocol was revised in 2018. Though the updated protocol recommended screening of women between 25 and 29 years with cytology every 3 years and 5-yearly HPV test followed by cytology triage for women between 30 and 65 years, cytology was still the most commonly available screening test in 2019–2020. Eligible participants are invited for cervical screening using the population lists maintained with general practitioners, primary health centres or insurance companies (*IARC, 2023c*). National participation to the cervical cancer screening programme is around 70% (*ICO/IARC, 2021b*; *WHO, 2021b* ).

Breast cancer screening is performed with clinical breast examination (CBE) for women aged 40–49 years, and with mammography for women aged 50–69 years. Only those women not screened in a particular round are sent a written invitation.

Colombia has an insurance-based system with insurance coverage of 99%. Thus, all preventive services must be provided by insurance companies and they have individual records for every service provided to their afiliates. The information system maintained by the insurance companies is used to manage and monitor the screening programme.

First case of COVID-19 was detected on 6 March 2020 in Colombia. As of 25 January 2023, there have been 6.3 million confirmed cases of COVID-19 with 142,385 deaths reported (0.65% of population) (*WHO, 2020d*). Health authorities imposed the initial lockdown from 24 March to 1 September 2020 throughout the country. Containment and other restrictions continued in several regions even after the lockdown was withdrawn.

Overall, a 46.3% reduction in the number of women screened with cervical cytology was observed in 2020 compared to 546,756 women screened in 2019 (*Table 2a*). The drop was most significant during the prolonged lockdown period with a slow recovery after that (*Figure 1b*).

The number of colposcopies performed in 2020 was 38.2% less compared to the same in 2019. During the initial lockdown period the drop exceeded 70%. The proportion of the screened women undergoing colposcopy in 2020 (2.9%) was similar to that in 2019 (2.5%) or 2018 (3.1%; data not shown in the table) and remained almost constant across the months. The reduction in the number of colposcopies was essentially because of a smaller number of women being screened during pandemic and not due to too many screen-positive women missing their further assessment.

The number of screening mammography performed in 2020 was 35.2% lower than the 96,940 examinations performed in 2019 (*Table 2b*). Highest reduction was observed immediately after the announcement of lockdown (85.8% in April). The volume of screening per month regained the pre-pandemic level almost immediately after withdrawal of lockdown. The total number of breast biopsies performed in 2020 was only 9.2% lower than that observed in 2019, since the programme could rapidly catch up with the missing biopsies during the lockdown. Proportion of biopsies among the total number of women undergoing mammography remained the same in 2020 (4.1%) as compared to 2019 (2.9%) and 2018 (2.7%; data not shown in table), signifying that the rate of further assessment of screen positives was maintained even during the pandemic months.

## Case study 3: Sri Lanka (high HDI category)

Sri Lankan ministry of health introduced breast and cervical cancer screening programmes in the years 1996 and 1998, respectively. For both breast and cervical cancer screening, women are invited through population registers to undergo screening at the age of 35 and a repeat screen at the age 45 (10 years later). Breast cancer sceening is carried out using CBE, while Pap-smear test is used for cervical cancer screening. National screening coverage for cervical cancer is around 29.3% (*IARC,*

**Table 2.** Monthly change in number of women undergoing cytology screening and colposcopy in the national cervical cancer screening programme in Colombia during 2019 and 2020 (A), and monthly change in number of women undergoing mammography screening and biopsies taken in the national breast cancer screening programme in Colombia during 2019 and 2020 (B).

**A** Monthly change in number of women undergoing cytology screening and colposcopy in the national cervical cancer screening programme in Colombia during 2019 and 2020

| Year of screening | Jan | Feb | Mar | Apr | May | Jun | Jul | Aug | Sep | Oct | Nov | Dec | Total |
|---|---|---|---|---|---|---|---|---|---|---|---|---|---|
| *Number of women screened (with cytology)* | | | | | | | | | | | | | |
| 2019 – n | 40,561 | 52,636 | 47,734 | 45,274 | 47,878 | 48,200 | 44,987 | 46,379 | 49,125 | 41,662 | 45,783 | 36,537 | 546,756 |
| 2020 – n | 49,680 | 47,205 | 26,564 | 14,510 | 6652 | 14,911 | 16,482 | 15,712 | 22,567 | 24,593 | 27,911 | 26,878 | 293,665 |
| Change – %* | +22.5 | −10.3 | −44.3 | −68.0 | −86.1 | −69.1 | −63.4 | −66.1 | −54.1 | −41.0 | −39.0 | −26.4 | −46.3 |
| *Number of women undergoing colposcopy* | | | | | | | | | | | | | |
| 2019 – n | 1614 | 1450 | 1222 | 1237 | 1249 | 998 | 1120 | 1182 | 1086 | 1114 | 1030 | 495 | 13,797 |
| 2020 – n | 1235 | 1165 | 897 | 350 | 485 | 579 | 623 | 684 | 631 | 656 | 697 | 528 | 8530 |
| Change – %* | −23.4 | −19.7 | −26.6 | −71.7 | −61.2 | −42.0 | −44.4 | −42.1 | −41.9 | −41.1 | −32.3 | +6.7 | −38.2 |

**B** Monthly change in number of women undergoing mammography screening and biopsies taken in the national breast cancer screening programme in Colombia during 2019 and 2020

| Year of screening | Jan | Feb | Mar | Apr | May | Jun | Jul | Aug | Sep | Oct | Nov | Dec | Total |
|---|---|---|---|---|---|---|---|---|---|---|---|---|---|
| *Number of women screened (with mammography)* | | | | | | | | | | | | | |
| 2019 – n | 9406 | 9050 | 7955 | 8880 | 8908 | 7966 | 8174 | 8880 | 8647 | 7779 | 6849 | 4446 | 96,940 |
| 2020 – n | 7837 | 9818 | 6855 | 1259 | 2269 | 3196 | 3483 | 3739 | 5125 | 6341 | 7189 | 5731 | 62,842 |
| Change – %† | −16.7 | 8.5 | −13.8 | −85.8 | −74.5 | −59.9 | −57.4 | −57.9 | −40.7 | −18.5 | +5.0 | +28.9 | −35.2 |
| *Number of women from whom biopsies were taken* | | | | | | | | | | | | | |
| 2019 – n | 184 | 311 | 255 | 203 | 287 | 228 | 213 | 223 | 226 | 253 | 257 | 194 | 2834 |
| 2020 – n | 211 | 302 | 273 | 98 | 136 | 148 | 201 | 207 | 208 | 264 | 270 | 256 | 2574 |
| Change – %† | +14.7 | −2.9 | −7.1 | −51.7 | −52.6 | −35.1 | −5.6 | −7.2 | −8.0 | +4.3 | +5.0 | +32.0 | −9.2 |

*Change is calculated as (numbers in 2020 − numbers in 2019)/(numbers in 2019) × 100.

†Change is calculated as (numbers in 2020 − numbers in 2019)/(numbers in 2019) × 100.

*2023d*). Provincial governments are responsible to carry out screening at Well Woman Clinics and send data to the national authorities through an Electronic Reproductive Health Management Information System (eRHMIS). Data related to cervical cancer screening at about 1000 clinic centres in 354 public health units in 28 health districts in the country are reported to the eRHMIS on a monthly basis.

First case of COVID-19 was detected in the country on 27 January 2020. Till 25 January 2023, 1.5 million confirmed cases of COVID-19 and 16,826 deaths were reported (0.04% of the population) (*WHO, 2020e*). Health authorities imposed nationwide lockdown for a brief period between 20 March and 24 March 2020. Partial lockdowns continued in high-prevalence zones.

Compared to 160,938 women screened with cytology in 2019, there was an overall 20.5% reduction in 2020 (*Table 3a*). Maximum reduction of 97% was documented in March 2020. The screening activities recovered quickly but went down again towards the year end as the number of COVID-19 cases surged (*Figure 1c*). The number of CIN 2+ detected in 2020 was 35.6% lower than that in 2019 with some reduction in detection rate in 2020 (0.7% of the screened women) compared to 2019 (0.9%) and 2018 (1.0%; data not shown in table).

The pattern of changes in breast cancer screening volume in 2020 mimicked changes in cervical screening volume in the same year with a short lasting but significant drop after the national lockdown (*Table 3b*). The number went down again at the year end due to surge in number of COVID-19 cases. Information on further assessment of the screen-positive women was not available.

## Case study 4: Thailand (high HDI category)

The Ministry of Public Health in cooperation with the National Health Security Office introduced breast, cervical, and colorectal cancer screening programmes in the years 2014, 2005, and 2018, respectively. Breast cancer screening is performed with CBE. Women between 30 and 60 years of age undergo Pap-smear in urban settings or VIA test in rural settings every 5 years (*IARC, 2023e*). HPV test was introduced in 2020 to replace cytology progressively. National cervical screening coverage was estimated to be around 67% in 2020 (*ICO/IARC, 2021c*; *Ploysawang et al., 2021*; *WHO, 2021c*). Colorectal cancer screening is based on the faecal immunochemical test (FIT) targeting men and women between 50 and 70 years every 2 years.

Thailand has a centralized health information system designed and maintained by Health Data Center (HDC) of Ministry of Health. Each primary care unit sends CRC screening data to HDC that links the colonoscopy data from the hospitals. However, the linkage between the hospitals and the information system is not yet complete. As a result, information on colonoscopy was not available for the individuals screened in 2019 and 2020.

Thailand was the first country to report a case outside China on 13 January 2020. As of 25 January 2023, 4.7 million confirmed cases of COVID-19 and 33,836 deaths were reported (0.07% of the population) (*WHO, 2020f*). Health authorities enforced lockdown between 22 March and 30 April 2020.

Data were obtained from colorectal cancer screening programme only. Compared to 832,394 individuals screened in 2019 for colorectal cancer there was an overall 30.7% drop in 2020 (*Table 4*). The programme managed to screen much higher number of individuals during September to December 2020 compared to same months in the year 2019 as more effort was given to scale up screening in the rural areas where the prevalence of COVID was less compared to the urban areas.

## Case study 5: Bangladesh (medium HDI category)

Bangladesh scaled up the opportunistic breast and cervical cancer screening programmes with revised protocols and policies in 2016. For breast cancer screening, CBE is performed on women attending various health facilities starting at 30 years of age. Visual inspection after application of acetic acid (VIA) is the cervical screening test offered to women aged between 30 and 60 years every 5 years (*IARC, 2023a*). The screen-positive women are referred for colposcopy at designated centres here treatment is performed based on colposcopic suspicion of high-grade lesions. A customized version of DHIS2 (a free and open-source software platform to collect aggregate and individual-level data) is used to collect data at district and national level, which permitted the analysis of COVID-19 (*Basu et al., 2021b*). However, information on further assessment of the CBE-positive women was lacking as the information system is yet to be extended to the centres where the women are referred for triple assessment. Till December 2022, 4.2 million were screened with VIA and 3.8 million women screened with CBE once in a lifetime in the programme. Estimated national coverage of the cervical and breast

**Table 3.** Monthly change in number of women undergoing cytology screening and high-grade CIN detected in the national cervical cancer screening programme in Sri Lanka during 2019 and 2020 (A), and monthly change in number of women undergoing CBE screening in the national cervical cancer screening programme in Sri Lanka during 2019 and 2020 (B).

**A** Monthly change in number of women undergoing cytology screening and high-grade CIN detected in the national cervical cancer screening programme in Sri Lanka during 2019 and 2020

| Year of screening | Month of screening | | | | | | | | | | | | Total |
|---|---|---|---|---|---|---|---|---|---|---|---|---|---|
| | Jan | Feb | Mar | Apr | May | Jun | Jul | Aug | Sep | Oct | Nov | Dec | |
| *Number of women screened (with cytology)* | | | | | | | | | | | | | |
| 2019 – n | 15,470 | 21,135 | 19,656 | 7467 | 12,352 | 13,038 | 11,309 | 11,055 | 11,590 | 11,797 | 11,044 | 15,025 | 160,938 |
| 2020 – n | 18,293 | 23,313 | 8149 | 222 | 1947 | 9650 | 14,519 | 12,543 | 12,153 | 7658 | 6748 | 12,812 | 128,007 |
| Change – %* | +18.2 | +10.3 | –58.5 | –97.0 | –84.2 | –26.0 | 28.4 | +13.5 | +4.9 | –35.1 | –38.9 | –14.7 | –20.5 |
| *Number of women detected with CIN 2 or worse lesions* | | | | | | | | | | | | | |
| 2019 – n | 28 | 10 | 9 | 2 | 6 | 10 | 11 | 16 | 13 | 13 | 7 | 24 | 149 |
| 2020 – n | 8 | 10 | 6 | 0 | 2 | 11 | 13 | 6 | 9 | 11 | 12 | 8 | 96 |
| Change – %* | –71.4 | 0.0 | –33.3 | –100.0 | –66.7 | +10.0 | 18.2 | –62.5 | –30.8 | –15.4 | +71.4 | –66.7 | –35.6 |

**B** Monthly change in number of women undergoing CBE screening in the national cervical cancer screening programme in Sri Lanka during 2019 and 2020

| Year of screening | Month of screening | | | | | | | | | | | | Total |
|---|---|---|---|---|---|---|---|---|---|---|---|---|---|
| | Jan | Feb | Mar | Apr | May | Jun | Jul | Aug | Sep | Oct | Nov | Dec | |
| *Number of women screened (with CBE)* | | | | | | | | | | | | | |
| 2019 – n | 19,972 | 27,150 | 26,329 | 10,288 | 16,643 | 17,778 | 17,947 | 15,890 | 16,884 | 17,667 | 16,207 | 20,253 | 223,008 |
| 2020 – n | 22,637 | 28,509 | 13,254 | 356 | 2303 | 12,271 | 17,982 | 15,563 | 15,966 | 10,057 | 8558 | 16,431 | 163,887 |
| Change – %† | +13.3 | +5.0 | –49.7 | –96.5 | –86.2 | –31.0 | +0.2 | –2.1 | –5.4 | –43.1 | –47.2 | –18.9 | –26.5 |

*Change is calculated as (numbers in 2020 – numbers in 2019)/(numbers in 2019) × 100; CIN: cervical intraepithelial neoplasia.
†Change is calculated as (numbers in 2020 – numbers in 2019)/(numbers in 2019) × 100; CBE: clinical breast examination.

**Table 4.** Monthly change in number of women undergoing screening with FIT in the national colorectal cancer screening programme in Thailand during 2019 and 2020.

| Year of screening | Month of screening | | | | | | | | | | | | Total |
|---|---|---|---|---|---|---|---|---|---|---|---|---|---|
| | Jan | Feb | Mar | Apr | May | Jun | Jul | Aug | Sep | Oct | Nov | Dec | |
| | *Number of women screened (with FIT)* | | | | | | | | | | | | |
| 2019 – n | 33,860 | 47,449 | 65,192 | 80,255 | 144,943 | 185,289 | 120,682 | 88,974 | 15,365 | 10,108 | 17,909 | 22,368 | 832,394 |
| 2020 – n | 27,076 | 33,731 | 62,283 | 23,556 | 38,778 | 88,622 | 93,219 | 70,560 | 61,219 | 24,049 | 22,549 | 31,608 | 577,250 |
| Change – %* | −20.0 | −28.9 | −4.5 | −70.6 | −73.2 | −52.2 | −22.8 | −20.7 | +298.4 | +137.9 | +25.9 | +41.3 | −30.7 |

*Change is calculated as (numbers in 2020 − numbers in 2019) / (numbers in 2019) ×× 100; FIT: faecal immunochemical test.

**Table 5.** Change in number of women undergoing VIA screening and colposcopy, and high-grade CIN detected in the national cervical cancer screening programme in Bangladesh during 2019 and 2020 (A), and change in number of women undergoing CBE and CBE positivity in the national breast cancer screening programme in Bangladesh during 2019 and 2020 (B).

| A | Change in number of women undergoing VIA screening and colposcopy, and high-grade CIN detected in the national cervical cancer screening programme in Bangladesh during 2019 and 2020 | | | | | | | | | | | | |
|---|---|---|---|---|---|---|---|---|---|---|---|---|---|
| Year of | Month of screening | | | | | | | | | | | | Total |
| screening | Jan | Feb | Mar | Apr | May | Jun | Jul | Aug | Sep | Oct | Nov | Dec | |
| | *Number of women screened (with VIA)* | | | | | | | | | | | | |
| 2019 – *n* | 34,132 | 31,448 | 39,775 | 33,559 | 21,434 | 26,694 | 33,366 | 27,597 | 35,636 | 37,019 | 36,208 | 34,663 | 391,531 |
| 2020 – *n* | 52,873 | 57,759 | 24,600 | 1719 | 2097 | 5078 | 11,740 | 30,644 | 30,250 | 33,466 | 47,093 | 39,088 | 336,407 |
| Change – %* | +54.9 | +83.7 | −38.2 | −94.9 | −90.2 | −81.0 | −64.8 | +11.0 | −15.1% | −9.6 | +30.1 | +12.8 | −14.1 |
| | *Number of women undergoing Colposcopy (data given for each quarter)*** | | | | | | | | | | | | |
| 2019 – *n* | 429 | | | 342 | | | 494 | | | 1265 | | | |
| 2020 – *n* | 440 | | | 86 | | | 393 | | | 919 | | | |
| Change – %* | +2.6 | | | −74.8 | | | −20.4 | | | −27.4 | | | |
| | *Number of women detected with CIN 2 or worse lesions (data given for each quarter)*** | | | | | | | | | | | | |
| 2019 – *n* | 106 | | | 62 | | | 75 | | | 243 | | | |
| 2020 – *n* | 56 | | | 38 | | | 68 | | | 162 | | | |
| Change – %* | −47.2 | | | −38.7 | | | −9.3 | | | −33.3 | | | |
| B | Change in number of women undergoing CBE and CBE positivity in the national breast cancer screening programme in Bangladesh during 2019 and 2020 | | | | | | | | | | | | |
| Year of | Month of screening | | | | | | | | | | | | Total |
| screening | Jan | Feb | Mar | Apr | May | Jun | Jul | Aug | Sep | Oct | Nov | Dec | |
| | *Number of women screened (with CBE)* | | | | | | | | | | | | |
| 2019 – *n* | 33,358 | 30,164 | 37,190 | 33,657 | 22,077 | 26,389 | 33,354 | 27,678 | 36,059 | 37,227 | 35,866 | 34,789 | 387,808 |
| 2020 – *n* | 51,313 | 57,271 | 24,508 | 1716 | 2046 | 5022 | 11,712 | 30,540 | 29,991 | 33,336 | 46,609 | 38,717 | 332,781 |
| Change – %[†] | +53.8 | +89.9 | −34.1 | −94.9 | −90.7 | −81.0 | −64.9 | +10.3 | −16.8 | −10.5 | +30.0 | +11.3 | −14.2 |
| | *Number and percentage of women who tested positive on CBE* | | | | | | | | | | | | |
| 2019 – *n* | 997 | 506 | 465 | 666 | 1048 | 1333 | 1202 | 875 | 856 | 1012 | 879 | 770 | 10,609 |
| (%) | (3.0) | (1.7) | (1.3) | (2.0) | (4.7) | (5.1) | (3.6) | (3.2) | (2.4) | (2.7) | (2.5) | (2.2) | (2.7) |
| 2020 – *n* | 594 | 778 | 234 | 37 | 63 | 99 | 114 | 158 | 382 | 272 | 950 | 367 | 4048 |
| (%) | (1.2) | (1.4) | (1.0) | (2.2) | (3.1) | (2.0) | (1.0) | (0.5) | (1.3) | (0.8) | (2.0) | (0.9) | (1.2) |

*Change is calculated as (numbers in 2020 − numbers in 2019)/(numbers in 2019) × 100; **Data presented are from the colposcopy clinic of the main hospital in Dhaka, Bangladesh; VIA: visual inspection with acetic acid; CIN: cervical intraepithelial neoplasia.

[†]Change is calculated as (numbers in 2020 − numbers in 2019)/(numbers in 2019) × 100; CBE: clinical breast examination.

cancer screening programme of the highly populous country still remains remains very low, around 5% only (**IARC, 2023b**; **ICO/IARC, 2021d** ).

The first known case of COVID-19 was confirmed on 8 March 2020. By 25 January 2023 there have been 2.0 million confirmed cases of COVID-19 with 29,441 deaths reported (0.02% of the population) (**WHO, 2020g**). The Government enforced a national lockdown from 26 March to 30 May 2020.

The cervical screening performance data presented in **Table 5a** show a significantly higher number of women being screened in the pre-pandemic months of 2020 due to introduction of a system of invitation by community health workers (CHWs) at least in selected districts (**Figure 1d**). Enlistment of the eligible women in the DHIS2 software supported this initiative. A significant reduction in the number of women being screened per month was observed during the lockdown and its immediate aftermath. The number of women screened per month quickly recovered and exceeded the previous year due to certain positive steps taken by the programme to continue with the screening activities

(e.g., ramping up screening in the rural areas less affected by COVID-19 and enforcing strict mitigation measures at the screening clinics).

Colposcopy data could be collected only from the largest colposcopy referral centre situated in the capital city of Dhaka. The number of colposcopies performed in 2020 was 27.4% less compared to that in 2019 with a maximum reduction of 74.8% being observed during the lockdown months. Number of CIN 2+ detected in 2020 showed an overall decrease of 33.3%.

The number of women being screened for breast cancer per month in 2019 and 2020 closely mimicked those for cervical cancer possibly because same women were subjected to both the tests (*Table 5b*). A 14.2% drop in overall number of women undergoing CBE was reported in 2020 compared to 2019 with the reduction being highest during the lockdown and a few months post-lockdown. Lower CBE positivity reported in 2020 compared to 2019 could be incidental as no definite pattern (no association with volume of screening, COVID-19-induced restrictions) was observed.

## Case study 6: Morocco (medium HDI category)

Ministry of Health of Morocco introduced breast and cervical cancer screening programme in the year 2010, which was gradually scaled up across all provinces. As per the breast cancer screening protocol updated in 2011, asymptomatic women aged from 40 to 69 years attending the PHCs are offered CBE every 2 years. VIA test is offered to women aged between 30 and 49 years every 3 years at the PHCs for cervical screening. The programme is opportunistic and supported by highly visible mass media campaigns (e.g., celebration of breast cancer awareness month throughout the month of October each year). Screen-positive women are referred to dedicated cancer diagnosis centres for further evaluation (*IARC, 2023f*). National screening coverage estimation for breast cancer is around 56.1% and for cervical cancer around 27.3% in the year 2019 (*IARC, 2019*).

Each screening centre and diagnostic centre maintains an electronic database of the women examined. The databases are not yet linked. Aggregate data are shared periodically with the national coordinator for preparing a performance report.

The first case of COVID-19 was detected on 2 March 2020 in Morocco. As of 25 January 2023, there have been 1.2 million confirmed cases of COVID-19 with 16,296 deaths reported (0.02% of the population) (*WHO, 2020h*). Health authorities imposed a nationwide lockdown between 20 March and 15 June 2020. Quarantines, partial lockdowns, and other restrictions continued even after that.

Almost all the regions of the country registered a significant drop in the volume of cervical screening in 2020 compared to the previous year resulting a new 55.2% reduction in the number of women screened in 2020 (*Table 6a*). The drop in screening activities was more in the urban provinces (e.g., 68.2% in Rabat, 61.6% in Fès/Meknès, and 58.5% in Grand Casablanca/Settat) compared to the semi-urban ones. The number of colposcopies performed in 2020 was 52.2% less compared to the number of colposcopies in 2019. However, the proportion of screened women undergoing colposcopy remained same in 2020 (2.8%) as compared to 2019 (2.6%) and 2018 (2.1%; data not shown on in table). By cascade effect, the number of CIN 2+ detected in 2020 was reduced by 20.7%.

The volume of breast cancer screening in 2020 (*N* = 766,702) reduced to almost half of that in 2019 (*N* = 1,515,604) with a higher impact registered in the urban provinces. A significant drop in further assessment was documented in 2020, with a decrease in number of mammography by 37.9% compared to the previous year (*Table 6b*). There was a 19.1% reduction in the number of breast cancers detected in 2020 compared to 2019. Breast cancer detection rate remained unchanged (1/1000 women screened in 2019 and 1.4/1000 women screened in 2020) indicating that the programme could provide appropriate further assessment services for the screen-positive women.

## Discussion

Breast, cervical, and colorectal cancer screening programmes achieving high coverage of the eligible population, as well as providing timely and high-quality diagnostic follow-up services can save significant number of lives. Despite all the complexities associated with implementation of such programmes and their high resource implications, many LMICs have heavily invested in rolling out cancer screening at population levels with varying amount of success. The COVID-19 outbreak forced most health systems to halt or slow down 'non-urgent' services to reassign staff for COVID-19-related work and reduce footfalls to the health facilities. Unfortunately, cancer screening has been included among such

**Table 6.** Site-specific change in number of women undergoing VIA screening and/or colposcopy, and high-grade CIN detected in the national cervical cancer screening programme in Morocco during 2019 and 2020 (A), and site-specific change in number of women undergoing CBE screening and/or mammography investigations, and breast cancers detected in the national breast cancer screening programme in Morocco during 2019 and 2020 (B).

**A**

Site-specific change in number of women undergoing VIA screening and/or colposcopy, and high-grade CIN detected in the national cervical cancer screening programme in Morocco during 2019 and 2020

| Year of screening | | Screening site | | | | | | | | | | Total |
|---|---|---|---|---|---|---|---|---|---|---|---|---|
| | Tanger Tetouan Al Hoceima | Fès Meknès | Rabat Salé Kénitra | Béni Mellal Khénifra | Grand Casablanca Settat | Marrakech Safi | Drâa Tafilalet | Souss Massa | Guelmim Oued Noun | Laayoune Sakia El Hamra | Eddakhla Oued Eddahab | |
| **Number of women screened (with VIA)** | | | | | | | | | | | | |
| 2019 – n | 10,965 | 115,348 | 45,904 | 15,777 | 72,186 | 46,113 | 8457 | 22,723 | 2521 | 1180 | 30 | 341,204 |
| 2020 – n | 4988 | 44,301 | 14,596 | 17,838 | 29,952 | 20,324 | 4387 | 13,350 | 2437 | 824 | 0 | 152,997 |
| Change – %* | −54.5 | −61.6 | −68.2 | 13.1 | −58.5 | −55.9 | −48.1 | −41.2 | −3.3 | −30.2 | −100.0 | −55.2 |
| **Number of women undergoing colposcopy** | | | | | | | | | | | | |
| 2019 – n | 899 | 2954 | 1520 | 832 | 1479 | 340 | 244 | 726 | | | | 8994 |
| 2020 – n | 303 | 1375 | 865 | 913 | 409 | 105 | 80 | 251 | | | | 4301 |
| Change – %* | −66.3 | −53.5 | −43.1 | 9.7 | −72.3 | −69.1 | −67.2 | −65.4 | | | | −52.2 |
| **Number of women detected with CIN 2 or worse lesions** | | | | | | | | | | | | |
| 2019 – n | 39 | 43 | 26 | 20 | 35 | 2 | 4 | 15 | | | | 184 |
| 2020 – n | 19 | 34 | 18 | 43 | 14 | 0 | 4 | 14 | | | | 146 |
| Change – %* | −51.3 | −20.9 | −30.8 | 115.0 | −60.0 | −100.0 | 0.0 | −6.7 | | | | −20.7 |

**B**

Site-specific change in number of women undergoing CBE screening and/or mammography investigations, and breast cancers detected in the national breast cancer screening programme in Morocco during 2019 and 2020

| Year of screening | | Screening site | | | | | | | | | | | Total |
|---|---|---|---|---|---|---|---|---|---|---|---|---|---|
| | Tanger Tetouan Al Hoceima | Oriental | Fès Meknès | Rabat Salé Kénitra | Béni Mellal Khénifra | Grand Casablanca Settat | Marrakech Safi | Drâa Tafilalet | Souss Massa | Guelmim Oued Noun | Laayoune Sakia El Hamra | Eddakhla Oued Eddahab | |
| **Number of women screened (with CBE)** | | | | | | | | | | | | | |
| 2019 – n | 113,755 | 99,230 | 269,517 | 210,892 | 158,305 | 249,501 | 190,757 | 45,150 | 150,947 | 18,569 | 8826 | 155 | 1,515,604 |
| 2020 – n | 60,170 | 76,848 | 99,917 | 81,357 | 101,285 | 117,061 | 99,441 | 23,521 | 85,295 | 13,648 | 8047 | 112 | 766,702 |
| Change – %† | −47.1 | −22.6 | −62.9 | −61.4 | −36.0 | −53.1 | −47.9 | −47.9 | −43.5 | −26.5 | −8.8 | −27.7 | −49.4 |
| **Number and percentage of women who tested positive on CBE** | | | | | | | | | | | | | |
| 2019 – n | 3908 | 805 | 4956 | 2888 | 2102 | 2572 | 1543 | 632 | 2324 | | | | 21,730 |
| (%) | (3.4) | (0.8) | (1.8) | (1.4) | (1.3) | (1.0) | (0.8) | (1.4) | (1.5) | | | | (1.4) |
| 2020 – n | 2807 | 418 | 3265 | 1563 | 1674 | 1354 | 976 | 316 | 1244 | | | | 13,617 |
| (%) | (4.7) | (0.5) | (3.3) | (1.9) | (1.7) | (1.2) | (1.0) | (1.3) | (1.5) | | | | (1.8) |
| **Number undergoing mammography investigations among CBE-positive women** | | | | | | | | | | | | | |
| 2019 – n | 1772 | 403 | 3506 | 2037 | 1378 | 2489 | 845 | 470 | 1311 | | | | 14,211 |
| 2020 – n | 1381 | 269 | 2267 | 982 | 709 | 1334 | 652 | 294 | 936 | | | | 8824 |

*Table 6 continued on next page*

*Table 6 continued*

**B** Site-specific change in number of women undergoing CBE screening and/or mammography investigations, and breast cancers detected in the national breast cancer screening programme in Morocco during 2019 and 2020

| | | | | | | | | | | |
|---|---|---|---|---|---|---|---|---|---|---|
| Change – %† | −22.1 | −33.3 | −35.3 | −51.8 | −48.5 | −46.4 | −22.8 | −37.4 | −28.6 | −37.9 |
| *Number of women detected with breast cancers* | | | | | | | | | | |
| 2019 – n | 249 | 82 | 346 | 186 | 65 | 204 | 79 | 49 | 115 | 1375 |
| 2020 – n | 213 | 16 | 271 | 174 | 65 | 125 | 91 | 30 | 127 | 1112 |
| Change – %† | −14.5 | −80.5 | −21.7 | −6.5 | 0.0 | −38.7 | 15.2 | −38.8 | 10.4 | −19.1 |

*Change is calculated as (numbers in 2020 −— numbers in 2019) / (numbers in 2019) x.x × 100; VIA: visual inspection with acetic acid; CIN: cervical intraepithelial neoplasia.

†Change is calculated as (numbers in 2020 − numbers in 2019)/(numbers in 2019) × 100; CBE: clinical breast examination.

'non-urgent' services in most countries (**WHO, 2020i**). The consequent reduction in cancer screening test volumes would result in substantial rise in avoidable cancer deaths (**Maringe et al., 2020**). Overwhelming of health systems due to the pandemic and associated detrimental consequences for the wider health and well-being of the population are likely to affect the countries with limited resources even worse than the high-income countries. However, little effort has been made to quantify the impact of the pandemic on cancer screening activities in the LMICs. This critical knowledge gap is generally ascribed to the absence of a system of collecting performance data on a regular basis to monitor the screening programmes in these countries.

Through CanScreen5 project of IARC we could have access to cancer screening performance data from selected LMICs. We categorized the LMICs by HDI to investigate any relationship between HDI and impact of the pandemic on screening performance. The high HDI countries having better healthcare systems are expected to have more resilient screening programmes. However, if MoHs interrupted preventive health services, then the possibility of continuing the screening is not related to the resilience of screening programmes. This was prossibly the reason why we did not see any specific patterns in the impact of COVID on cancer screening performance by HDI.

In an ongoing and stable cancer screening programme the monthly or yearly volume of screening tests usually remains similar between consecutive years, as has been demonstrated by the comparison of cervical screening data between 2018 and 2019 in the countries included in our study (**Figure 1**). The drop in the number of monthly or yearly screening volumes reported in our study for 2020 provides a reliable estimate of the number of individuals that missed screening due to the pandemic. Irrespective of the HDI category, every country observed the biggest drop in screening during the lockdown. In almost all countries the first lockdown declared in 2020 was invariably at a short notice and was highly disruptive to routine healthcare. The health systems did not get enough time and opportunities to reorganize non-emergency services like cancer screening. The screen-eligible individuals were very much hesitant to visit health facilities due the fear of getting infected. As the lockdowns were withdrawn and the health systems adopted several measures to get adjusted to the new situation, the volume of screening in most of the programmes gradually returned back to the pre-pandemic level, as demonstrated in our study. Nonetheless, during the intervening period thousands of individuals have missed their screening, which will have a significant impact on cancer mortality in due course. What is more worrying is our observation that many of the screen-positive individuals in some of the countries missed their further assessment and as a result there were fewer detection of precancers and cancers among them.

The reduction in the test volumes in 2020 compared to previous year observed in the LMICs was quite comparable to that observed in high-income countries (e.g., cancer screening test volumes were reduced by 40% overall in 2020 in cancer screening programmes in Ontario, Canada) (**Walker et al., 2021**). Some of the LMIC programmes demonstrated remarkable resilience by catching up with the unscreened eligible individuals, at least partially, within a short period of time. This included both high HDI countries (e.g., breast cancer screening in Colombia, colorectal cancer screening in Thailand) and medium HDI countries (e.g., breast and cervical cancer screening in Bangladesh). In addition to creating 'safe' facilities for the screen-eligible individuals, Bangladesh and Thailand expanded the screening activities to the rural and hard-to-reach areas where the COVID-19 prevalence was not high due to low population density (**Basu et al., 2021b**). Despite experiencing a significant drop in the volume of screening in 2020, the breast and cervical screening programmes in Colombia and Morocco managed to ensure further assessment of the screen-positive women essentially by strengthening the referral pathways (e.g., having dedicated cancer diagnostic clinics outside the routine health facilities for the screen-positive women to attend).

Cancer screening and further assessment of screen-positive individuals were significantly impacted by the pandemic induced service discontinuation, containments, and travel restrictions by all programmes irrespective of their settings. Such disruptions may happen in future as well, due to not only natural calamities but also political turmoil. Since all countries are trying to 'build back' their health systems better, attention should be given to make them more resilient and equipped to prevent, diagnose, and provide care for non-communicable diseases including cancer. Unlike COVID-19, cancer is a pandemic that is going to stay and likely to get worse with time. By introducing pragmatic protocols for screening (e.g., less frequent screening with more sensitive tests, self-collection of samples preferably at home), further assessment (e.g., molecular tests to triage self-collected samples)

and treatment (e.g., screen and treat for cervical precancers), and use of technologies like telemedicine for consultation or mobile application for delivery of test results the programmes can reduce the necessity for the screen-eligible individuals to visit health facilities too frequently (*Basu et al., 2021a*). Learning lessons from each other is crucial.

We have produced the rarely available quantitative data on the change in screening volumes during COVID pandemic in selected LMICs. Modelers may use the data to estimate the additional number of cases that would be detected in advanced stage or excess deaths that are expected, as has been done in several high-income countries (*Castanon et al., 2020*; *Yong et al., 2021*). The study also highlights the importance of CanScreen5 project of IARC that aims to build capacities across the globe to systematically collect data across cancer screening continuum and use the same for programme management and quality improvement (*Zhang et al., 2023*).

Our study has several limitations. Many of the programmes could not provide data across the screening continuum (screening, further assessment, detection of disease and treatment), which precluded a complete assessment of the pandemic impact. This was essentially because most LMIC programmes do not have an effective health information system to collect data across screening pathways for programme monitoring and quality assurance. There may be several reasons that may decrease or increase the volume of screening in a particular year and could have impacted our estimates of COVID-19 impact. The number of individuals being screened is usually high with the launch of a new programme or initiation of a new round of screening. This happened with the colorectal cancer screening programme in Thailand in our study as they initiated a new round in 2020. In a regularly screened population, the number of individuals screened may come down during the end of a particular screening round. However, this was not the case in any of the programmes included in our study. Many of the programmes used screening tests well known to have highly variable performance due to their subjective nature – for example, VIA for cervical screening and CBE for breast cancer screening. Unpredictable variation in the test positivity (as has been demonstrated for CBE screening in Bagladesh and Morocco) across settings and comparison periods could have influenced the downstream management, and explain the observed variations in rates of further assessment or disease detection.

To conclude, our analysis demonstrates the profound impact of COVID-19 pandemic on cancer screening services irrespective of HDI category of the countries. It is important for all programmes to strategize how to ramp up services to clear the backlogs in screening and more crucially in further evaluation of screen positives. Following the example of the countries reporting data in the present study, having a robust information system to collect cancer screening data is imperative to monitor screening-related service volumes and backlogs. Well-coordinated, decisive, and collective actions remain critical to make screening programmes more equitable and resilient in the face of natural and geo-political calamities.

## Acknowledgements

Where authors are identified as personnel of the International Agency for Research on Cancer/World Health Organization, the authors alone are responsible for the views expressed in this article and they do not necessarily represent the decisions, policy or views of the International Agency for Research on Cancer /World Health Organization.

## Additional information

### Funding
No external funding was received for this work.

### Author contributions
Eric Lucas, Formal analysis, Writing - original draft, Writing – review and editing; Raul Murillo, Silvina Arrossi, Martin Bárcena, Youssef Chami, Ashrafun Nessa, Suraj Perera, Padmaka Silva, Suleeporn Sangrajrang, Data curation, Writing – review and editing; Richard Muwonge, Formal analysis, Writing – review and editing; Partha Basu, Conceptualization, Supervision, Methodology, Writing - original draft, Writing – review and editing

## Author ORCIDs

Eric Lucas [ORCID] http://orcid.org/0000-0002-1252-1925
Raul Murillo [ORCID] http://orcid.org/0000-0001-7187-9946
Partha Basu [ORCID] http://orcid.org/0000-0003-0124-4050

## Decision letter and Author response

Decision letter https://doi.org/10.7554/eLife.86527.sa1
Author response https://doi.org/10.7554/eLife.86527.sa2

## Additional files

### Supplementary files

• MDAR checklist

### Data availability

All data generated or analysed during this study are included in the manuscript.

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
