## [Editor Report]

This study provides important estimates from an international cancer screening data repository about the impact of the COVID-pandemic related disruptions on cancer screening programs in selected low- and middle-income countries. The evidence supporting the study is solid and relies on national-level screening program attendee volumes and assessments of screen positives during 2019 (pre-pandemic) and 2020 (during the pandemic). The study provides real-world data estimates of proportions/volumes of missed screenings due to pandemic control measures (lockdowns and closures) and may contribute to future modelling efforts for measuring the impact on late/advanced stage detection and excess case burden and mortality.

---

## [Decision Letter]

**Decision letter after peer review:**

Thank you for submitting your article "Quantification of impact of COVID-19 pandemic on cancer screening programmes -a case study from Argentina, Bangladesh, Colombia, Morocco, Sri Lanka and Thailand" for consideration by *eLife*. Your article has been reviewed by 3 peer reviewers, and I oversaw the evaluation in my dual role of Reviewing Editor and Senior Editor. The following individuals involved in the review of your submission have agreed to reveal their identities: Linda Rabeneck (Reviewer #2); Mona Saraiya (Reviewer #3).

As is customary in *eLife*, the reviewers have discussed their critiques with one another and with the Reviewing and Senior Editors. The decision was reached by consensus. What follows below is an edited compilation of the essential and ancillary points provided by reviewers in their critiques and in their interaction post-review. Please submit a revised version that addresses these concerns directly. Although we expect that you will address these comments in your response letter, we also need to see the corresponding revision clearly marked in the text of the manuscript. Some of the reviewers' comments may seem to be simple queries or challenges that do not prompt revisions to the text. Please keep in mind, however, that readers may have the same perspective as the reviewers. Therefore, it is essential that you amend or expand the text to clarify the narrative accordingly.

*Reviewer #1 (Recommendations for the authors):*

Suggest including the weakness related to the high variability of performance of screening tests (especially those with subjective interpretation such as VIA for cervical cancer or clinical breast exam) across the comparison periods such that screen positivity rates may have been affected in unpredictable ways.

Figure 1 could be provided as a higher-resolution image.

*Reviewer #2 (Recommendations for the authors):*

1) I would pare down the Discussion section a bit, as in some places it is a bit speculative.

2) In the Discussion section you indicate an intention to investigate any relationship between HDI and the impact of Covid on screening performance. But truthfully, you can't really do this, as you have acknowledged, hence the case study (rather than a comparative) design. So I would tone this down a bit.

3) In the Discussion section you indicate that the aim of CRC screening is to detect precancerous lesions…giving more time to catch up on screening without a major impact on cancer incidence. This is a rather bold statement…not sure I would agree. CRC screening with a quantitative FIT screening detects some precancerous lesions, depending on the cut-off chosen. gFOBT does not detect precancerous lesions. Colonoscopy detects precancerous lesions, but it is not the most commonly used CRC screening test outside of the US.

4) In the Discussion section, the statement with respect to breast screening and early detection…that the Covid-related decrease in screening would have a greater impact on breast cancer mortality needs a reference.

5) In the Discussion section, I would make a point about the usefulness of CanScreen5 to conduct this type of study. It is an important resource for the field in my view.

Nice work!

*Reviewer #3 (Recommendations for the authors):*

This paper examines the impact of COVID on screening programs for countries outside of EUROPE /North America, which is valuable and well done in the current format.

The paper does an excellent job describing the situation, the tables and figures are very informative, and the authors do a good job describing limitations and strengths.

Here are a few suggestions:

Please review some grammar/parallel sentence structure (perhaps the editor can help)

Data are vs. Data is.

The abstract mentions the importance of having data from LMICs-but most of the countries are High HDI or Medium HDI-but to be consistent, use the same verbiage that is noted in the introduction of the main paper-and not imply you have a low-income country (or low HDI). Just that you are examining countries.

It would be helpful to know why you were not able to obtain other screening program data from countries that have breast and cx (e.g. Thailand).

In all figures and tables, please document currently if figures don't represent the country but just a province (e.g. Argentina) or specific clinic (this was done for Bangladesh). This can be done through footnoting.

Use of work opportunistic is used for Bangladesh and Morrocco. Was that purposeful? Should there be any mention of whether the other countries are not considered opportunistic considered "organized"?

A table of the countries and relevant info about them would be good to document, but also concern about comparing each other.

It will be good for authors to estimate if they will be able to measure mortality or incidence impact (stage) in these countries or not.

In tables, for the reader not familiar with clinical sites (any info on whether a clinic is rural or urban (e.g. Morocco table 6))?

You may want to mention the role of electronic health reporting in most of these countries and how that's come along considerably, in some cases, this kind of information cannot even be obtained at a country level in the US, for example.

---

## [Author Response]

Essential revisions:Reviewer #1 (Recommendations for the authors):Suggest including the weakness related to the high variability of performance of screening tests (especially those with subjective interpretation such as VIA for cervical cancer or clinical breast exam) across the comparison periods such that screen positivity rates may have been affected in unpredictable ways.Figure 1 could be provided as a higher-resolution image.

We thank the reviewer for encouraging comments. We have added the following to ‘weakness’ section in the discussion.

“Many of the programmes used screening tests well known to have highly variable performance due to their subjective nature – e.g., VIA for cervical screening and CBE for breast cancer screening. Unpredictable variation in the test positivity (as has been demonstrated for CBE screening in Bangladesh and Morocco) across settings and comparison periods could have influenced the downstream management and explain the observed variations in rates of further assessment or disease detection.”

We will submit a high-resolution image of figure 1.

Reviewer #2 (Recommendations for the authors):1) I would pare down the Discussion section a bit, as in some places it is a bit speculative.2) In the Discussion section you indicate an intention to investigate any relationship between HDI and the impact of Covid on screening performance. But truthfully, you can't really do this, as you have acknowledged, hence the case study (rather than a comparative) design. So I would tone this down a bit.3) In the Discussion section you indicate that the aim of CRC screening is to detect precancerous lesions…giving more time to catch up on screening without a major impact on cancer incidence. This is a rather bold statement…not sure I would agree. CRC screening with a quantitative FIT screening detects some precancerous lesions, depending on the cut-off chosen. gFOBT does not detect precancerous lesions. Colonoscopy detects precancerous lesions, but it is not the most commonly used CRC screening test outside of the US.4) In the Discussion section, the statement with respect to breast screening and early detection…that the Covid-related decrease in screening would have a greater impact on breast cancer mortality needs a reference.5) In the Discussion section, I would make a point about the usefulness of CanScreen5 to conduct this type of study. It is an important resource for the field in my view.Nice work!

We thank the reviewer for very useful comments and kind words of appreciation.

We have shortened the paragraph 2 of discussion that dealt with the relation between COVID and HDI and removed the paragraph 4 that dealt with different impact by cancer sites.Indeed, we didn’t get suitable references to support the statement in para 4.We have added reference to CanScreen5 (discussion; para 5), the first report of which will soon be published in Nature Medicine.

Reviewer #3 (Recommendations for the authors):This paper examines the impact of COVID on screening programs for countries outside of EUROPE /North America, which is valuable and well done in the current format.The paper does an excellent job describing the situation, the tables and figures are very informative, and the authors do a good job describing limitations and strengths.Here are a few suggestions:Please review some grammar/parallel sentence structure (perhaps the editor can help)Data are vs. Data is.

We have reviewed the manuscript to make appropriate grammatical changes and language corrections.

The abstract mentions the importance of having data from LMICs-but most of the countries are High HDI or Medium HDI-but to be consistent, use the same verbiage that is noted in the introduction of the main paper-and not imply you have a low-income country (or low HDI). Just that you are examining countries.

We have made changes in the abstract to reflect that the data was collected from high and medium HDI countries only.

It would be helpful to know why you were not able to obtain other screening program data from countries that have breast and cx (e.g. Thailand).

Screening programmes for different cancer sites are often managed by different institutions/teams in countries and the quality of data depends on programme organization and maturity. In Thailand breast cancer screening data were not available essentially due to poor organization. Thailand switched to HPV detection-based screening in many regions just before the pandemic. The programme managers didn’t want to share data from a programme that was still being rolled out and data collection was not yet complete. We didn’t want to include such information in the manuscript as that could hurt the sentiments of the programme managers.

In all figures and tables, please document currently if figures don't represent the country but just a province (e.g. Argentina) or specific clinic (this was done for Bangladesh). This can be done through footnoting.

We have changed the figure 1 legend to reflect that the programme in Argentina was only regional. The fact that the colposcopy data from Bangladesh was only from a clinic has already been mentioned in the footnote of table 5a.

Use of work opportunistic is used for Bangladesh and Morrocco. Was that purposeful? Should there be any mention of whether the other countries are not considered opportunistic considered "organized"?

We have indicated for each study site whether the programme was opportunistic or population-based. In addition to Bangladesh and Morocco, the Argentinian programme was also primarily opportunistic.

A table of the countries and relevant info about them would be good to document, but also concern about comparing each other.

We have described programme organization in the case studies. We don’t want to add more tables. As you have rightly mentioned, comparison of organization of the programmes was not our intention.

It will be good for authors to estimate if they will be able to measure mortality or incidence impact (stage) in these countries or not.

There is a team at IARC that is working on modelling the impact of delayed cancer diagnosis during the pandemic on cancer-specific mortality. We have not yet had any explicit discussion with them about using cancer screening data.

In tables, for the reader not familiar with clinical sites (any info on whether a clinic is rural or urban (e.g. Morocco table 6))?

Provinces in Morocco cannot be classified rural or urban because most of the regions are nixed of rural and urban.

You may want to mention the role of electronic health reporting in most of these countries and how that's come along considerably, in some cases, this kind of information cannot even be obtained at a country level in the US, for example.

We have added to each case study a paragraph on the information system used to collect data as this will indeed be of interest to the readers.